# Transcriptome Profiling of Phenylalanine-Treated Human Neuronal Model: Spotlight on Neurite Impairment and Synaptic Connectivity

**DOI:** 10.3390/ijms251810019

**Published:** 2024-09-18

**Authors:** Sara Stankovic, Andrijana Lazic, Marina Parezanovic, Milena Stevanovic, Sonja Pavlovic, Maja Stojiljkovic, Kristel Klaassen

**Affiliations:** 1Institute of Molecular Genetics and Genetic Engineering, University of Belgrade, Vojvode Stepe 444a, 11042 Belgrade, Serbia; sstankovic@imgge.bg.ac.rs (S.S.); andrijanak@imgge.bg.ac.rs (A.L.); marina.parezanovic@imgge.bg.ac.rs (M.P.); milenastevanovic@imgge.bg.ac.rs (M.S.); sonya@imgge.bg.ac.rs (S.P.); maja.stojiljkovic@imgge.bg.ac.rs (M.S.); 2Institute of Physiology and Biochemistry “Ivan Djaja”, Faculty of Biology, University of Belgrade, Studentski trg 16, 11158 Belgrade, Serbia; 3Serbian Academy of Sciences and Arts, Kneza Mihaila 35, 11001 Belgrade, Serbia

**Keywords:** phenylketonuria, RNA sequencing, transcriptome, gene expression profile, NT2-derived neurons, neurite impairment, synaptic connectivity

## Abstract

Phenylketonuria (PKU) is the most common inherited disorder of amino acid metabolism, characterized by high levels of phenylalanine (Phe) in the blood and brain, leading to cognitive impairment without treatment. Nevertheless, Phe-mediated brain dysfunction is not fully understood. The objective of this study was to address gene expression alterations due to excessive Phe exposure in the human neuronal model and provide molecular advances in PKU pathophysiology. Hence, we performed NT2/D1 differentiation in culture, and, for the first time, we used Phe-treated NT2-derived neurons (NT2/N) as a novel model for Phe-mediated neuronal impairment. NT2/N were treated with 1.25 mM, 2.5 mM, 5 mM, 10 mM, and 30 mM Phe and subjected to whole-mRNA short-read sequencing. Differentially expressed genes (DEGs) were analyzed and enrichment analysis was performed. Under three different Phe concentrations (2.5 mM, 5 mM, and 10 mM), DEGs pointed to the *PREX1*, *LRP4*, *CDC42BPG*, *GPR50*, *PRMT8*, *RASGRF2,* and *CDH6* genes, placing them in the context of PKU for the first time. Enriched processes included dendrite and axon impairment, synaptic transmission, and membrane assembly. In contrast to these groups, the 30 mM Phe treatment group clearly represented the neurotoxicity of Phe, exhibiting enrichment in apoptotic pathways. In conclusion, we established NT2/N as a novel model for Phe-mediated neuronal dysfunction and outlined the Phe-induced gene expression changes resulting in neurite impairment and altered synaptic connectivity.

## 1. Introduction

Phenylketonuria (PKU, OMIM #261600) is the most common inborn error of amino acid metabolism with an average prevalence of 1:10,000 newborns in Europe [1,2]. It is one of the most studied rare diseases and the first one with newborn screening introduced [2]. This autosomal recessive disorder is caused by pathogenic variants in the gene coding for hepatic enzyme phenylalanine hydroxylase (PAH). PAH catalyzes the conversion of L-phenylalanine (Phe) to tyrosine (Tyr), requiring tetrahydrobiopterin (BH4) as a cofactor, along with iron and molecular oxygen [3,4]. Variants in the *PAH* gene affect the structure and/or function of this enzyme leading to its complete or partial inability to perform the conversion [5,6]. The result is the accumulation of phenylalanine in the blood and, after crossing the blood–brain barrier, in the brain [6].

Untreated PKU patients develop severe intellectual disability, the primary clinical feature of PKU, along with epilepsy, psychiatric and behavioral issues, and movement abnormalities [3,7]. To prevent neurological and cognitive deficits, the early initiation of dietary treatment is necessary for PKU patients. Nevertheless, even in patients who adhered to the recommended dietary regimen, the negative effects on executive function, attention, memory, and mood remain a concern during adolescence and adulthood [1]. Despite its effectiveness, the Phe-restricted diet does not provide an optimal solution for every treatment goal, especially for adult patients with emerging neurological manifestations, thus opening the door to novel therapeutic approaches [1]. On the other side of the spectrum, rare “unusual” cases of late-diagnosed, never-treated PKU patients without severe intellectual disability despite high plasma Phe concentrations have been reported [8,9,10]. Interestingly, although these patients had almost normal intellectual functioning, many of them showed other cerebral PKU symptoms, and, in some of them, neurological symptoms started later in life. These interesting findings further imply the importance of understanding the molecular pathophysiology of PKU.

Despite extensive research on the molecular pathophysiology of PKU over the past decades, the exact cause of brain dysfunction mediated by high Phe levels remains incompletely understood. Many mechanisms of Phe’s influence on the brain have been suggested [1]. In mouse cultured neurons, it has been previously shown that hyperphenylalaninemia disrupts dendritic outgrowth and synaptic connectivity [11,12]. It has also been suggested that energy production is impaired in the frontal cortex of hyperphenylalaninemic animal models as well as in PET images of PKU patients [1,13]. Additionally, the hyperphenylalaninemic state has been associated with the disrupted influx of long neutral amino acids (LNAAs), along with neurotransmitter deficiency. While all these cerebral effects align with cognitive and neurodevelopmental issues in PKU patients, the precise molecular mechanism of how Phe damages neurons remains unclear [1]. It is likely that many different mechanisms are involved in the brain pathophysiology of PKU [1]. The Phe-mediated brain dysfunction continues to be a topic of great interest for scientists, highlighting the complexity of the neuropathology underlying this seemingly well-known monogenic rare disease.

The origins of PKU brain pathophysiology research involved animal models, leading to the first main findings such as the importance of neurotransmitter metabolism impairment and defects in protein synthesis, with mice and rat models as the most widely used [14]. The first genetic PKU model and by far the most commonly used one is the Pah^enu2^ mouse model, resembling the genetic and biochemical characteristics, as well as the neurobiology, of human PKU, and providing the first evidence of the direct neurotoxic effect of Phe on the brain. The biggest drawback with this model is the lack of comparability when it comes to cognitive impairment [14,15]. Nevertheless, animal models became vital in studies of the gene expression alterations and transcriptome profile of PKU brain neuropathology, a powerful novel technique with promising results. A study conducted on the brain tissue of PKU mice subjected to either a low Phe diet or a standard diet revealed differential gene expression between the two groups, which could be partially reversed by the low Phe diet. Additionally, the study identified distinct gene expression alterations in two examined brain regions: the hippocampus and the cerebral cortex [16]. Common methods for gene expression alterations include RT-PCR or microarray analysis [17,18,19], but, recently, RNA sequencing (RNA-seq) has emerged as a method of choice in transcriptome studies, with advantages such as the direct sequencing of RNA molecules, greater dynamic range for detection, and less background noise [20]. A rather small number of manuscripts featuring RNA-seq in PKU were found upon a PubMed search encompassing papers from January 2014 to July 2024. Two studies performed RNA-seq on animal models, one on the avian model of maternal PKU [21], the other with RNA isolated from Pahenu2 mouse liver [22]. However, the gene expression data for PKU patients or model systems obtained by RNA-seq are still scarce. Recently, Kim et al. conducted RNA sequencing on iPSC-derived cerebral organoids treated with phenylalanine, observing differences in organoid size, apoptosis, and the depletion of progenitor cells [23]. 3D brain organoids were proposed as a new model for PKU research with the hope of having a better resemblance with the brain complexity in humans and a chance to explore differences between patients by deriving specific organoids from their iPSCs [24]. Along with their indisputable advantages, it should be noted that the organoid phenotype most closely resembles the prenatal brain (in terms of structure, function, and transcriptome), and it poses issues with variability and reproducibility, along with the costly and time-consuming development [24,25]. On the other hand, the classic 2D cell culture of rat and mouse models are also widely used, but the weak analogy with the human brain and the lack of cell–cell interactions remain the leading drawbacks of these models [26,27]. In one study, human neuroblastoma cell lines were treated with Phe to assess its effect on neural cell proliferation [28]; however, to the best of our knowledge, no studies concerning the effect of Phe were carried out on human neuron-like cells.

The NTERA-2 (NT2) cell line is derived from human embryonal testicular carcinoma cells. NT2 cells are human-based and have a unique potential to differentiate into both neurons and glial cells [29]. They closely resemble and exhibit many features of human embryonic neural stem cells, while the process of differentiation into neurons resembles vertebrate neurogenesis [29,30]. The differentiation of NT2 cells to NT2-derived neurons (NT2/N) is irreversible and the acquired cells strongly resemble human neurons in morphology and functionality [29,31]. In NT2/N cultures, functional synapses have been identified, with neurotransmitters and neurotransmitter receptors detected and action potentials generated on depolarization [29,32]. NT2/N cells are widely accepted as a suitable model for the study of neuronal networks and synaptogenesis processes [33]. NT2/N cells have been compared to primary human neurons and showed a clear resemblance with the same ratio of neurite to cell body area, despite the slight size difference [29]. These cells have proven to be a valid alternative source of terminally differentiated neurons and a reliable source of neurons for studying different neurological diseases and the screening of novel neuroprotective therapeutics [29]. NT2/N cells have been widely accepted as an experimental model to study disease pathophysiology as well, such as APP protein metabolism in Alzheimer’s disease or mitochondrial function in Parkinson’s disease [34,35]. Moreover, differentiating NT2 cells and NT2/N cells can be used for screening neurochemicals and understanding the cellular mechanisms of their effects [30,36].

In this study, NT2/N cells were differentiated from NT2/D1 cells and treated with various Phe concentrations with the aim to assess Phe-mediated neuron dysfunction. This is the first study that strives to evaluate the effect of Phe on gene expression in mature neuronal model, the NT2/N cells. We hypothesized that Phe treatment would lead to changes in the gene expression of NT2/N cells compared to untreated, control neurons. The excessive Phe exposure of the mature neuronal model followed by short-read mRNA sequencing could enable the detection of not only the Phe-induced differentially expressed genes but also the overrepresented gene sets and biological pathways implicated in neuronal dysfunction. Our main goal was to elucidate the mechanisms behind neuronal impairment caused by Phe, the crucial component of the PKU cognitive phenotype.

## 2. Results

### 2.1. Characterization of NT2/D1 Cell Differentiation into NT2/N In Vitro

The retinoic-acid-induced differentiation of human pluripotent embryonal carcinoma NT2/D1 cells in vitro was achieved as established in the literature [37,38]. The induction of NT2/D1 cells with RA causes a shift in marker expression from neuroepithelial to neuronal, yielding NT2/N. We verified the mature neuronal phenotype of NT2/N by observing the expression of specific cytoskeletal proteins. Microtubule-associated protein 2 (MAP2), a neuron-specific cytoskeletal protein present throughout neuronal cell bodies and processes, has shown intense immunostaining, by which we confirmed the mature neuronal state of the differentiated NT2/N (Figure 1).

### 2.2. Phenylalanine Treatment of NT2/N and Cell Viability Assay

NT2/N cells plated on a 96-well plate were treated with different concentrations of Phe and their viability was analyzed after 7 days of treatment using the 3-(4,5-dimethylthiazol-2-yl)-2,5-diphenyltetrazolium bromide (MTT) assay. Phe treatments with concentrations ranging from 3 mM to 30 mM led to a significant decrease in cell viability. We detected a survival rate ranging from 57.8% to 1.3% following the gradient of concentrations, while the Phe concentration at which 50% of cells survived was 5.45 mM (Appendix A). We observed that Phe significantly affected cell viability at all concentrations used, but most prominently at the 24 mM and 30 mM concentrations (cell survival < 3%, *p*-value < 0.005) (Appendix A). Cell survival had decreased to 57.8% after 7 days upon treatment with 3 mM Phe. Therefore, for our subsequent NT2/N treatment, we selected one physiological concentration with cell survival over 75% (1.25 mM Phe), three higher concentrations distributed around the 50% cell survival point, with an estimated viability between 35% and 65% (2.5 mM, 5 mM, and 10 mM), as well as one extremely high nonphysiological concentration (30 mM Phe).

### 2.3. RNA Sequencing Quality Control and Mapping to the Reference Genome

In this study, we used the total RNA from NT2-derived neurons treated with five different concentrations of Phe (1.25 mM; 2.5 mM; 5 mM; 10 mM; and 30 mM) and untreated NT2/N controls for cDNA library preparation. Eighteen samples in total (biological triplicates of each concentration treatment and of untreated controls) were subjected to short-read mRNA-sequencing. Firstly, quality control was performed to assess the RNA sequencing accuracy and reliability. The sequencing error rate and GC content distribution were determined for each sample, as well as the raw read and mapping quality assessment and reads distribution in the reference genome. One sample treated with 1.25 mM Phe failed the quality control and was excluded from further analysis, leaving a biological duplicate for the lowest Phe concentration. The base calling accuracy, represented by Q scores, showed that the quality of the sequencing data was sufficient for further analysis, with Q20 and Q30 over 95% and 90% for each sample, respectively. Additionally, the GC content varied from 50.51% to 52.80%. The aligning of clean reads against the Hg38 Homo sapiens reference genome revealed that over 91% of reads were mapped for each sample. Uniquely mapped reads accounted for 88.38% to 92.81% of the total mapped reads. Overall, 17 samples had satisfactory quality measures and were used for further expression analysis.

### 2.4. Co-Expressed and Uniquely Expressed Genes

The estimation of the gene expression level of the tested samples was performed by counting the reads mapped to the reference genome. The FPKM (fragments per kilobase of transcript sequence per millions base pairs sequenced) value for each gene was calculated based on the length of the gene and sequencing depth (average reads count mapped to the gene) and used as the final parameter of gene expression. We used Venn diagrams for the visualization of the co-expressed and uniquely expressed genes for all Phe concentrations in comparison to untreated cells (Figure 2). We observed a markedly lower number of uniquely expressed genes in 1.25 mM-Phe-treated cells (89) compared to cells treated with higher Phe concentrations (468, 236, 690, and 3993 for 2.5 mM, 5 mM, 10 mM, and 30 mM Phe treatment, respectively) (Figure 2A). The co-expression of genes was particularly prevalent among groups treated with 2.5 mM, 5 mM, and 10 mM Phe, with a total of 114 genes co-expressed across all three groups (Figure 2B).

### 2.5. Principal Component Analysis

In order to identify data clustering, we performed principal component analysis (PCA). The PCA results pointed out two groups that separated clearly, as shown on the scatterplot (Figure 3). We detected a group of samples treated with 30 mM Phe clustered together, highlighting a distinct expression pattern in comparison to the other treatment groups and untreated NT2/N. All treatments except the highest one (30 mM) are found in close proximity with untreated cells after PCA visualization (Figure 3), indicating that their expression profiles did not vary markedly. We performed one-way ANOVA followed by post hoc tests to validate the clustering observed in the PCA plot. The group treated with 30 mM Phe showed a statistically significant difference in comparison to the 2.5 mM, 5 mM, and 10 mM treatment groups, while a statistical trend was observed in comparison to the control cells (corrected *p*-value = 0.06). No other groups showed any statistically significant differences. However, the small sample size might introduce bias. Upon batch effect correction, no major clustering changes were observed on the PCA plot, which was confirmed after statistical analysis (one-way ANOVA, followed by post hoc tests). Consequently, we concluded that the batch effect did not contribute significantly to our results. Overall, PCA suggested that the main difference among expression data is found between the samples treated with 30 mM Phe and the rest of the samples, explaining around 85% of the data variability (Top 10 PC1 and PC2 loadings visualized in Appendix A).

### 2.6. Analysis of Differentially Expressed Genes

We examined differentially expressed genes (DEGs), comparing each Phe treatment group with untreated cells using the following cutoff significance parameters: absolute log_2_FoldChange > 0.5 and adjusted *p*-value < 0.05 (Appendix A). The total numbers of DEGs that passed the aforementioned criteria were as follows: 5 for the 2.5 mM Phe group (2 up-regulated and 3 down-regulated; Figure 4A), 7 for the 5 mM Phe group (5 up-regulated and 2 down-regulated; Figure 4B), 55 for the 10 mM Phe group (40 up-regulated and 15 down-regulated; Figure 4C), and 1285 for the 30 mM Phe group (433 up-regulated and 852 down-regulated; Figure 4D). We did not detect DEGs meeting the cutoff criteria in the 1.25 mM Phe treatment group. The lists of all significant DEGs in the four Phe treatment groups, as well as the full gene names, are available in Appendix A.

The expression profiles we observed across groups treated with 2.5 mM, 5 mM, and 10 mM Phe revealed distinct expression trends, with significant co-expression among DEGs. The genes *PREX1*, *LRP4*, *CDH6,* and *BMP5* had a statistically significant overexpression under both the 5 mM and 10 mM Phe treatment, while 10 more DEGs (among which are *IGFBP3*, *PCSK9,* and *PRMT8*) had a statistically significant differential expression under the 10 mM Phe treatment and a statistical trend observed at the 5 mM Phe concentration. We identified 24 DEGs that were significantly differentially expressed in the 10 mM Phe group while also showing statistical significance or a statistical trend at two lower concentrations, 2.5 mM and 5 mM Phe. Among these genes were the following: *CDH6*, *SOX3*, *CDC42BPG*, *LRP4*, *PREX1*, *AEBP1*, *GPR50,* and *RASGRF2*. To confirm the gene expression alterations in the 2.5 mM, 5 mM, and 10 mM Phe treatment groups, we performed qRT-PCR for the following genes: *CDC42BPG*, *CDH6*, *GPR50*, *LRP4*, *NTNG2*, *PCSK9*, *PREX1*, *PRMT8,* and *RASGRF2*. Statistically significant down-regulation was observed for *PRMT8*, in accordance with transcriptome DEGs analysis, while all other selected genes showed statistically significant up-regulation, supporting the transcriptome findings (Appendix A). 

Among the identified DEGs, many have distinctive roles in central nervous system (CNS) functioning and a connection to neuronal physiology and diseases (Table 1). Some of the genes detected as DEGs have a role in cytoskeletal dynamics alteration and regulation, including *PREX1* and *LRP4*. A significant portion of DEGs was also connected to neurite morphology and functionality regulation, such as *PREX1*, *LRP4*, *CDC42BPG*, *GPR50*, *CDH6,* and *PRMT8*. We detected that genes encoding synaptic organizers and other proteins associated with synaptic transmission and plasticity to have an altered expression in at least one of the groups treated with 2.5 mM, 5 mM, and 10 mM Phe (*LRP4*, *RASGRF2*, *CDH6*, and *PRMT8*). Three differentially expressed genes highlighted after Phe treatments are known to be involved in GTP/GDP exchange and G-coupled protein regulation in the brain (*PREX1*, *GPR50,* and *RASGRF2*). Many of the identified DEGs are also considered significant in connection to different diseases that affect the CNS such as Alzheimer’s disease, amyotrophic lateral sclerosis (ALS), Parkinson’s disease, bipolar disorder, and depression (such as *LRP4*, *GPR50,* and *PCSK9*), as well as clinical features such as intellectual disability and epilepsy (*CDC42BPG*, and *LRP4*). Additionally, among the DEGs identified in this study, *PCSK9*, *KLF4,* and *MT1E* have roles including apoptosis and neuroinflammation.

The treatment of NT2/N with the highest, nonphysiological Phe concentration (30 mM) has shown conflicting results. The expression pattern of DEGs identified in the 30 mM Phe treatment group did not support the findings observed at lower concentrations. The total sum of differential genes observed after a comparison of the 30 mM Phe treatment with control was markedly higher than in other treatment groups, with uniquely-expressed genes accounting for about 87% of the total sum. The data for the 30 mM Phe treatment group have shown not only a drastically increased number of DEGs (1285) compared to other concentrations, but also more down-regulated genes, in contrast with our findings at the 5 mM and 10 mM Phe concentration. In the 30 mM Phe treatment group, we did not observe a statistically significant differential expression for any of the genes that had an altered expression at the 2.5 mM, 5 mM, and 10 mM Phe treatment and that were previously mentioned as important for neuron functioning, pointing out the distinct effects that Phe exerts on neurons in extreme nonphysiological conditions.

### 2.7. Gene Set Enrichment Analysis and Overrepresentation Analysis of DEGs

Gene set enrichment analysis (GSEA) was used to determine differences in gene sets of Phe-treated groups in comparison to the untreated group. Genes were ranked based on the log_2_FoldChange values. Gene ontology (GO) and Reactome terms overrepresented in gene sets revealed key differences in the biological processes, cellular components, and molecular functions in treated NT2/N.

We observed a significant portion of GO terms related to neuron projections and the cytoskeleton, such as neuron projection guidance, the positive regulation of cell projection organization, neuron projection extension involved in neuron projection guidance, the positive regulation of actin filament polymerization, and the cluster of actin-based cell projections under the 2.5 mM, 5 mM, and 10 mM Phe treatments (Figure 5A–C, and Appendix A). In the same treatment groups, we detected overrepresented terms connected to G protein-coupled receptor activity and signaling pathway. Synapse-related processes were abundant within the enriched gene sets under the 5 mM and 10 mM Phe treatment, among which we highlight the following: synaptic vesicle recycling, synaptic signaling, and neurotransmitter uptake (Figure 5B,C). In the 2.5 mM treatment group, a major portion of the overrepresented gene sets encompasses the respiratory chain complex and mitochondrial ATP synthesis (Figure 5A). The Wnt signaling pathway and protein serine/threonine kinase signaling pathway were enriched under treatments with 5 mM and 10 mM Phe (Figure 5B,C). The results from GSEA using the Reactome database showed similar overrepresented terms in comparison to GO (Appendix A), as well as additional ones, such as the metabolism of amino acids and G protein beta-gamma signaling.

The overrepresented gene sets that we observed in the 30 mM Phe treatment group after GSEA mostly did not correspond to the ones we detected at lower concentrations. Among the noted GO terms, many were connected to the response to external biotic stimulus and cell cycle regulation, as well as the regulation of cell migration and cell motility (Figure 5D). However, among the enriched terms, we also observed neurotransmitter loading into the synaptic vesicle and actin cytoskeleton (Figure 5D). Overrepresented Reactome terms included MAPK activation-related pathways and the release cycle for neurotransmitters such as dopamine, glutamine, norepinephrine, and serotonin (Appendix A).

Additionally, in search of significantly enriched genes and pathways, we performed an overrepresentation analysis (ORA) for the 10 mM and 30 mM Phe treatments (since these two groups showed more than 50 DEGs) using the Gene Ontology, Kyoto Encyclopedia of Genes and Genomes (KEGG), and Reactome databases. Among the enriched DEGs under 10 mM Phe treatment, most were found in two distinct groups of GO terms in relation to neuron projections and synapse organization (Appendix A). Terms connected to neuronal processes and overrepresented within DEGs include the regulation of neuron projection development and arborization, as well as the regulation of dendrite development (Appendix A). Another group that stood out after GO enrichment analysis included the following synaptic processes: synaptic membrane adhesion, presynapse and postsynapse assembly, and presynapse and postsynapse membrane organization. Other GO and Reactome terms that emerged as significant in the 10 mM Phe group were as follows: small-GTPase-mediated signal transduction, and the regulation of actin-filament-based processes and cytoskeleton organization (Appendix A). Another Reactome term that emerged as significantly overrepresented was diseases of the neuronal system (Appendix A).

The GO enrichment analysis that we performed for the 30 mM Phe treatment revealed various overrepresented terms, among which are cell migration, the ERK1 and ERK2 cascade, and the cellular response to biotic stimulus (Appendix A). Enriched GO terms we observed at the highest Phe concentration treatment that are involved in neuronal functioning included neuron projection guidance, the response to axon injury, and the regulation of actin filament organization (Appendix A). KEGG enrichment analysis was used to identify significantly overrepresented biological pathways corresponding to detected DEGs in the 30 mM Phe treatment group (Appendix A). KEGG pathways that were identified as significantly enriched mostly matched the GO terms observed as overrepresented. Among the overrepresented biological pathways were the p53 signaling pathway, cellular senescence, the MAPK signaling pathway, axon guidance, and the regulation of the actin cytoskeleton (Appendix A). The results obtained using Reactome supported the ones observed with GO and KEGG (Appendix A).

## 3. Discussion

PKU is a well-studied and long-known rare disease, yet the precise mechanisms by which Phe impairs neurons are still not completely elucidated, occupying the attention of scientists for decades [1]. In the years behind us, many Phe-mediated effects have been recognized and researched, but the precise mechanisms and gene expression alterations remain unclear [1]. The most widely recognized mechanisms of phenylalanine’s impact on the brain include white matter impairment, the disruption of energy production and metabolism, the influx of LNAA, and neurotransmitter deficiency [1]. The aim of our study was to shed light on gene expression alterations, as they could point out potential mechanisms underlying morphological and physiological neuron dysfunction and facilitate future research on elucidating PKU neuropathology. 

In this study, we investigated the gene expression changes following phenylalanine treatment in mature human-based neurons derived from NT2/D1 cells. Upon induction with RA, NT2/D1 cells differentiate into NT2/N, CNS-like mature neurons that generate action potentials and calcium spikes, and express neurofilaments, as well as release and respond to neurotransmitters [23]. An important advantage of the NT2/D1 cell line is its differentiation into terminally differentiated neurons with the complete loss of tumor characteristics [39,40]. Upon applying different Phe concentrations to the NT2/N cells, we evaluated its effect on gene expression using mRNA sequencing. The changes we observed have shown varying concentration-dependent gene expression patterns, as well as the identification of DEGs involved in various neuronal functions.

In order to precisely assess the Phe-mediated NT2/N gene expression alterations, we decided to use a set of increasing concentrations, encompassing the physiological ones, as well as the extreme ones. From the literature search, focusing on studies on mouse and rat cultured neurons [28,41], we opted for the 1.25 mM Phe concentration, which was in concordance with the results observed after the MTT assay showing the survival rate of NT2/N over 75%. Furthermore, we applied three higher Phe concentrations that were still in the range of substantial NT2/N survival (35–65%), 2.5 mM, 5 mM, and 10 mM, as used in previous studies on animal models [12,42]. Finally, an extreme concentration of 30 mM Phe was chosen to represent an acute high phenylalanine load [23]. We opted for 7 days of treatment as it was commonly used for NT2/N treatments [31] as well as in animal cultured neuron Phe treatments [42], providing enough time for detectable Phe-mediated changes in cell morphology and gene expression, while minimizing the probability of external factors contributing to neuronal cell death.

Interestingly, the treatments of NT2/N with 2.5 mM, 5 mM, and 10 mM Phe showed similar gene expression profiles and DEGs. This was further confirmed by qRT-PCR (Appendix A). The most striking observation is that exposure to the all applied Phe concentrations, 2.5 mM, 5 mM, and 10 mM, caused the differential expression of genes involved in two distinct processes: neurite impairment and synaptic connectivity. 

As per neurite impairment, we noted that several of the DEGs were connected to neuronal projections, such as the processes of neurite formation, growth, and arborization, as well as cytoskeleton dynamics alteration and regulation (Figure 6). Among the up-regulated genes, *PREX1*, *LRP4*, *CDC42BPG*, and *GPR50* stood out, while *PRMT8* was a notable down-regulated gene. The *PREX1* gene, which was significantly overexpressed under the 2.5 mM, 5 mM, and 10 mM Phe treatments, encodes a RAC3 guanine nucleotide exchange factor (GEF) that functions in neuronal cells, and it was shown to inhibit neurite elongation and modify actin cytoskeletal dynamics at the growth cone [43]. Rac-GEFs were identified as important players in the regulation of actin cytoskeletal organization, with opposing roles among the protein family members [44]. Additionally, the overexpression of *P-rex1* in the prefrontal cortex of mice resulted in abnormal neuronal polarity, migration, and dendritic spine morphology [45]. *LRP4* is another gene whose protein product is proposed to be a regulator of the cytoskeleton, explaining both its roles in the development of presynaptic terminals and the impairment of dendrite morphogenesis [46,47]. *LRP4* is well-known as a part of the neuromuscular junction, but, recently, its versatile roles in the brain have emerged [46]. The overexpression of *Lrp4* was shown to result in shorter but more numerous dendrites and decrease the mobility of dendrite terminals [47]. Two more up-regulated genes whose overexpression is shown to lead to dendrite changes are *CDC42BPG* and *GPR50*. *Cdc42bpg* is a modulator of dendritic spine morphogenesis and, when overexpressed in cortical neurons in culture, leads to a decrease in the mean length of dendritic protrusions in the mouse cerebellar model [48], while *Gpr50* was connected to increased filopodia- and lamellipodia-like structure formation [49]. Both of these genes are also involved in G-protein coupled receptor signaling. A gene that we observed to be down-regulated in our study, and appears to be exclusive to the brain [50], is *PRMT8*. This protein arginine methyltransferase is connected to many cellular processes including DNA reparation and signal transduction [51], and, even though research about gene expression alterations of this gene are scarce, there is one study showing that *Prmt8* homozygous knockout mice had neurons with reduced dendritic arborization and stunted dendritic trees [50]. To the best of our knowledge, none of these genes were previously mentioned in the context of PKU. However, the changes in their gene expression do match the neuron changes previously observed in animal models and cell culture [12,42,52]. The enriched processes and pathways obtained by GSEA for groups treated with 2.5 mM, 5 mM, and 10 mM Phe are in line with the gene expression alteration findings. Among the enriched terms are neuron projection morphogenesis and guidance, actin filament organization, and the cluster of actin-based cell projections. One study on rat cortical neurons in culture, using a similar Phe concentration range as ours (0.9 mM to 5 mM), had shown a significant decrease in the length of the longest neurites 24 h post-treatment [26]. Another study has shown that Phe interferes with dendritic arborization by affecting the neuronal actin cytoskeleton [42]. Additionally, the enriched GO terms observed in a study performed on 12-week-old brain tissue of PKU mice and wild-type mice on a normal diet included axon development and guidance [16], supporting our findings. To conclude, all of the mentioned gene expression and pathway changes are in line with the morphological and physiological changes observed in animal models and post-mortem human research, providing, for the first time, new knowledge on Phe-mediated neurite impairment.

Synapse formation and synaptic connectivity disturbances were connected to PKU pathophysiology in mouse models [53,54], and even a modifier gene family involved in synaptic transmission has been suggested in a study on untreated PKU patients without intellectual disability [8]. In our study, we have identified a subset of genes with known roles in synapse functioning that are statistically significant or showing a statistical trend in gene expression alterations in groups treated with 2.5 mM, 5 mM, and 10 mM Phe (Figure 6). Previously mentioned for its dendrite-related effects, *LRP4* also has a role in synaptic connectivity, supporting its potential function as a regulator of the cytoskeleton [48]. This gene has recently emerged as a synapse organizer in the brain, where it is enriched in membranes, suggesting both pre- and postsynaptic roles in the brain [12,22,55]. Another up-regulated gene in our study is *RASGRF2*, known for its role in calcium-mediated signaling, but also a significant role in synaptic potentiation [56]. This gene is predominantly expressed in the adult brain and involved in the conversion of calcium transients in synapse function [56,57]. The aforementioned *PRMT8* gene is a promising candidate as a regulator in synaptic function and could have a role in cognitive processes, while one study suggested its possible synapse membrane localization, since it is known that *PRMT8* can be membrane-bound [58,59]. Another observed differentially expressed gene, *CDH6*, was shown to enable NLGN1-dependent presynaptic differentiation [60]. As expected, based on DEGs, among the enriched GO terms, the Reactome terms and pathways obtained from GSEA were synaptic vesicle recycling, synaptic signaling, and neurotransmitter loading into the synaptic vesicle, while enrichment analysis for the 10 mM Phe group also showed presynaptic and postsynaptic membrane organization. Previously, Hong et al. have used a microarray to determine the gene expression differences of PKU and wild-type mice, and, among the significant GO terms, they found calcium ion binding, the modulation of chemical synaptic transmission, and the positive regulation of synapse assembly [16]. Our study supports the findings that synaptic connectivity, membrane assembly, and transmission are important contributing factors for PKU pathology, while addressing some new potential gene candidates involved in Phe-mediated gene expression changes. Furthermore, these findings, as well as previously mentioned findings related to axon development and guidance [16], confirm that Phe-treated NT2/N recapitulates findings observed in a well-established PKU model and therefore implies its usefulness as a novel human-cell-based PKU model.

Intriguingly, our analysis did not show many statistically significant DEGs, which could be explained by the fact that phenylalanine is an amino acid, naturally present in the cell, unlike chemical drug compounds, inorganic toxins, environmental exposures, etc. that, rather, provoke an altered gene expression response [61]. Therefore, it was not unexpected that the treatment with 1.25 mM Phe did not yield any DEGs. Comparably, in the study conducted by Dobrowolski et al. on DNA methylation in the pathophysiology of hyperphenylalaninemia in the PAH^enu2^ mouse model, promoter hypermethylation of only four genes in the PKU brain was discovered [62]. Additionally, a study conducted on gene expression in the brain tissue of wild-type PKU mice, PKU mice kept on a low-Phe diet for 12 weeks, and PKU mice kept on a standard diet for the same period of time revealed 23 DEGs between PKU mice on the standard diet and wild-type mice and 30 DEGs between mice on the standard and low-PKU diet [16]. These results are in concordance with our findings and the proposed explanation of fewer differentially expressed genes due to the natural presence of Phe in cells.

Having addressed the potential involvement of the identified DEGs and enriched gene sets in processes connected to PKU pathophysiology, we further investigated the association of the aforementioned genes to the pathology of other diseases accompanied by neuron impairment. Existing evidence of gene expression change in other diseases would strengthen the case of their potential contribution in the PKU cognitive phenotype. The previously described *GPR50* gene has been proposed as a contributor to the development of mental disorders [49], while *LRP4* was connected to ALS, Alzheimer’s disease, and epilepsy [63,64]. We also identified the overexpression of *PCSK9*, a gene encoding a member of the subtilisin-like proprotein convertase family, whose expression in the adult brain is normally low but is highly up-regulated during disease states [65]. This gene was connected to brain apoptosis with conflicting results showing both pro- and antiapoptotic functions, and it was proposed to work in the brain through both the intrinsic and extrinsic apoptotic pathways [65,66]. *PCSK9* might play a role in Alzheimer’s disease, but the exact mechanisms are yet to be elucidated [67]. Given the mentioned role of *CDC42BPG* on dendritic spine morphogenesis and a previously known case of another small GTPase-dependent kinase involved in cognitive problems, one study suggested that *CDC42BPG* could participate in the signaling pathways involved in intellectual disability and autism spectrum disorders [48]. Overall, many of the identified DEGs are considered significant in connection to different diseases that affect the CNS, providing new potential targets for PKU research and supporting the idea that multiple mechanisms are needed to explain the complex pathophysiology of the PKU cognitive phenotype.

Unlike common expression patterns visible in three different (2.5 mM, 5 mM, and 10 mM Phe) treatment groups, the group treated with an extremely high dose of Phe (30 mM) has shown a poor survival rate together with completely different DEGs and enriched gene sets. Besides having the most uniquely expressed genes and notably more down- than up-regulated genes, none of the previously mentioned genes with an altered gene expression and a potential role in neuronal functioning were noted in the 30 mM Phe group. Expectedly, GSEA has revealed some distinctive GO terms, such as cell cycle regulation, and the regulation of cell migration and cell motility, while GO enrichment analysis also revealed the ERK1 and ERK2 cascade as an overrepresented term. KEGG enrichment, as well as Reactome, pointed out important pathways such as the p53 signaling pathway and MAPK signaling pathway. We have shown that this nonphysiological Phe concentration has a neurotoxic effect and that the cell survival rate of NT2/N is extremely low after treatment with 30 mM Phe. In line with these observations, we expected to see a different expression profile after transcriptomic analysis and more apoptosis-related DEGs. Indeed, the p53 and MAPK signaling pathways, highlighted by KEGG enrichment analysis, are both involved in apoptosis and the response to stress. Among the overrepresented Reactome terms, neurotransmitter release cycles were also noted. The enriched GO terms connected to neurons included actin cytoskeleton reorganization and response to axon injury, both supporting the neurotoxicity of Phe. In PKU research, cells are rarely treated with these extreme Phe concentrations, since they do not represent physiological conditions. Thus, the literature data about the effect of these extreme doses of Phe are scarce. Recently, Kim et al. used an even higher Phe concentration (50 mM) to treat organoids and observed alterations in organoid size and the induction of apoptosis, while RNA-seq revealed the differential expression of genes related to apoptosis and the stress response, as well as the enriched p53 pathway [23]. Our gene expression results do support the findings of this study.

The limitation of our study is the rather short duration of phenylalanine exposure compared to the in vivo conditions. Future studies providing long-term exposure to Phe are needed to better reassemble its effects in vivo for a borderline classical PKU concentration. In addition to the rather short treatment period, another limitation of this study is that NT2/N cells were grown as a neuron-enriched culture. Growing the cells in a mixed culture with mature glial cells would be more suitable in terms of representing the complexity and interactions in human CNS. A further limitation of this study is the relatively small number of samples, which reduces the statistical power of the study. Future research with larger sample sizes would be beneficial for improving the reliability of the findings. Furthermore, this approach covers only the quantitative difference in gene expression patterns, so it could be useful to employ long-read sequencing to provide insight into novel transcripts in the future.

In line with our aim to elucidate gene expression patterns in PKU, for the first time, we proposed NT2/D1-derived neurons as a novel model for the Phe-mediated neuronal effect and, also for the first time, pointed out the *PREX1*, *LRP4*, *CDC42BPG*, *GPR50*, *PRMT8*, *RASGRF2,* and *CDH6* genes in the context of PKU. Among the advantages of this human-based model is NT2/N’s ability to form functional synapses, which makes it an appropriate model for studying neuronal networks and synapses [33], and a useful asset in our study. Despite NT2/N cells mostly being used in the context of researching their differentiation as a model of human neuronal differentiation, we have proposed that NT2/N can be a suitable PKU model for studying the effects of various molecular processes in mature neurons. In addition to analyzing mRNA, this model can be used to further study small non-coding RNA, long non-coding RNA, methylation processes, proteomes, or metabolomes, thus contributing to the overall understanding of disease pathology. Moreover, this cost-effective in vitro PKU model may be used for the preliminary testing of potential new RNA therapeutics or small-molecule compounds aiming to correct disturbed processes in the brain. 

Many observations about the Phe-mediated neuron effects have been recorded in animal models and led to important conclusions. This study, however, highlights the importance of corroborating these findings on human-based models, employing a simpler cellular NT2/N model. Overall, this study underscores the significance of gene expression alterations in response to the dose-dependent Phe treatment as a path towards further PKU studies and paves the way for novel therapeutics research and elucidating the molecular mechanisms underlying the complex PKU neuropathology.

## 4. Materials and Methods

### 4.1. NT2 Differentiation to NT2-Derived Neurons

The human embryonic teratocarcinoma cell line NT2/D1 (CRL-1973™, ATCC, Manassas, VA, USA) was cultured in high-glucose (4500 mg/L) Dulbecco’s Modified Eagle’s medium (DMEM) supplemented with 10% fetal bovine serum (FBS), and 2 mmol/L L-glutamine and antibiotic-antimycotic mixture (10,000 units/mL penicillin, 10,000 μg/mL streptomycin, and 25 μg/mL amphotericin B) (Thermo Fisher Scientific, Waltham, MA, USA), at 37 °C in a humidified atmosphere containing 10% CO_2_ as previously described [68]. To induce differentiation in culture, cells were treated with 10 μmol/L all-trans retinoic acid (RA; Sigma-Aldrich, Burlington, MA, USA) for 4 weeks. A neuron-enriched population was then isolated as described by Pleasure et al. [37,69]. Briefly, following RA induction, cells were trypsinized and plated at sixfold lower density. After 2 days in culture, neuron-like cells were detached from the plate by tapping mechanically on the side of the tissue culture plate and 1 × 10^6^ cells were re-plated on Geltrex^®^ (Thermo Fisher Scientific, Waltham, MA, USA)-coated petri dishes (ϕ 6 cm). Cells were grown for the following 7 days in the presence of the following mitotic inhibitors: 10 mmol/L uridine, 10 mmol/L 5-fluoro-5-deoxyuridine, and 1 mmol/L cytosine arabinoside (Sigma-Aldrich, Burlington, MA, USA).

### 4.2. Immunocytochemistry

NT2/N cells were plated on Geltrex^®^-coated cover slips and fixed in 4% paraformaldehyde in phosphate-buffered saline (PBS) for 20 minutes at room temperature (RT). Next, cell permeabilization was achieved using 0.2% Triton X-100 in PBS for 20 minutes and cells were blocked in 1% bovine serum albumin (BSA) and 5% normal goat serum in PBS with 0.1% Triton X-100 at RT for 1 h. The mouse primary antibody against microtubule-associated protein 2 (ab11267, Abcam, Cambridge, UK), a marker specific for mature neurons, was diluted in a ratio 1:500 in PBS containing 1% BSA and 0.1% Triton X-100. The samples were incubated at 4 °C overnight, and, afterwards, the cover slips were washed 3 times for 10 min in 0.1% Triton X-100 prepared in PBS. Next, samples were incubated with anti-mouse secondary antibody conjugated with Alexa Fluor^®^ 594 (Thermo Fisher Scientific, USA), diluted in PBS containing 1% BSA and 0.1% Triton X-100 (1:1500) for 1 h at RT. The nuclei were then stained with 0.1 mg/mL diamino phenylindole (DAPI; Sigma-Aldrich, Burlington, MA, USA) and coverslips were mounted with mounting medium (Thermo Fisher Scientific, Waltham, MA, USA). Samples were viewed under a Leica TCS SP8 confocal microscope with LAS AF-TCS SP8 software (version 3.1.2) used (Leica Microsystems, Wetzlar, Germany).

### 4.3. Cell Viability Assay

Following RA induction and 2 days in culture, neuron-like cells were seeded in 96-well plate at a density of 1.5 × 10^4^ cells per well. For the following 7 days, cells were cultured in the presence of mitotic inhibitors, as previously described. Five Phe-enriched media were made by dissolving Phe (Sigma-Aldrich, Burlington, MA, USA) in High-Glucose DMEM with 10% FBS and 1% antibiotic-antimycotic mixture (Thermo Fisher Scientific, Waltham, MA, USA) to achieve final concentration range of 3 mM to 30 mM. NT2/N were treated with 5 concentrations of Phe for 7 days, with medium changed every 2 days. Cell viability was tested on the 7th day using the MTT assay (Merck, Rahway, NJ, USA), according to the manufacturer’s instructions. Infinite 200 PRO microplate reader (Tecan, Männedorf, Switzerland) was used to obtain colorimetric measurements. The assay was carried out in four technical replicates and repeated in two independent experiments. The statistical significance between the groups treated with different Phe concentrations and untreated NT2/N was estimated by Student’s t-test. The results of the MTT assay analysis were visualized on a barplot using ggplot2 R package (version 4.4.1, R Core Team, Vienna, Austria) [70]. Concentration where cell viability was 50% was calculated and used to determine concentrations for subsequent Phe treatment, along with literature search.

### 4.4. Phenylalanine Treatment

After plating on Geltrex^®^ (Thermo Fisher Scientific, Waltham, MA, USA)-coated petri dishes, along with mitotic inhibitors, NT2/N were treated with 5 Phe concentrations, as follows: 1.25 mM; 2.5 mM; 5 mM; 10 mM; and 30 mM Phe. The treatment was administered for 7-days period in biological triplicates for each Phe concentration. Additionally, a triplicate of untreated NT2/N was maintained as a control. Phe (Sigma-Aldrich, Burlington, MA, USA) was dissolved in High-Glucose DMEM with 10% FBS and 1% antibiotic-antimycotic mixture (Thermo Fisher Scientific, Waltham, MA, USA) and NT2/N cells were treated by culturing in this Phe-enriched medium. Control cells were cultured in a medium with the same composition but lacking Phe. The medium was changed every 2 days for all treatment and control groups, and fresh mitotic inhibitors were added along with the new medium. Treatment doses were chosen so that the lower Phe concentrations simulate classic PKU (cPKU, >1200 μmol/L blood Phe concentration), whereas higher Phe concentrations were used to investigate severe effects of Phe on neurons.

### 4.5. RNA Extraction and mRNA Sequencing

On the 7th day of treatment, cells were collected. Briefly, monolayers of cells from 6 cm petri dishes were trypsinized and centrifuged (1500 rpm for 7 min), and the supernatant was carefully removed. Total RNA was isolated using a phenol-chloroform-based procedure, TRIzol Reagent (Ambion^®^, Thermo Fisher Scientific, Waltham, MA, USA), according to the manufacturer’s instructions. The quantity and purity of isolated RNA were measured using a BioSpec-nano spectrophotometer (Shimadzu, Kyoto, Japan), as well as Qubit 4 Fluorometer (Thermo Fisher Scientific, Waltham, MA, USA). Poly-T oligo-attached magnetic beads were used for messenger RNA (mRNA) purification from total RNA. The cDNA library was prepared and the total mRNA sequencing was performed using Illumina platform. The mRNA purification, cDNA library construction, and RNA sequencing were conducted by Novogene (UK) Company Limited, Cambridge, UK.

### 4.6. Quality Control, Mapping to Reference Genome, and Differential Expression Analysis

Raw data were first processed by quality control, trimming and removing adapter sequences. Clean reads were further obtained by removing low-quality reads and reads containing poly-N sequence. The quality parameters, Q20 and Q30, and GC content were calculated and the high-quality clean reads were used for downstream analysis. Clean reads were aligned to the human reference genome (GRCh38.p14) using Hisat2 v2.0.5 [71]. The number of reads mapped to each gene was determined by featureCounts v1.5.0-p3 [72], and the expected number of fragments per kilobase of transcript sequence per millions base pairs sequenced of each gene was calculated based on the gene length and sequencing depth. FPKM was used for estimating the gene expression levels. Furthermore, principal component analysis was performed on variance-stabilized raw counts data with the top 500 most variable genes among samples using DESeq2 [73] R package (version 4.4.1, R Core Team, Vienna, Austria). Additionally, we performed batch effect correction using sva [74] R package (version 4.4.1, R Core Team, Vienna, Austria), as our NT2/N were obtained from 2 separate differentiations. PCA results were visualized as a scatterplot, using ggplot2 [70] R package (version 4.4.1, R Core Team, Vienna, Austria). Differential expression analysis and statistical analysis was performed using the DESeq2 R package (version 4.4.1, R Core Team, Vienna, Austria) [75]. Furthermore, the resulting *p*-values were adjusted using the Benjamini and Hochberg’s method [76], in order to minimize the false discovery rate. The genes expressed with an absolute log_2_FoldChange > 0.5 and adjusted *p*-value < 0.05 were considered as significant DEGs.

### 4.7. Gene Set Enrichment Analysis and Overrepresentation Analysis

Gene set enrichment analysis was performed to determine gene sets showing a significant consistent difference between treatment groups and control. Gene Ontology [77] and Reactome databases [78] were used and the analysis was accomplished with clusterProfiler R package (version 4.4.1, R Core Team, Vienna, Austria) [79,80]. The genes were ranked according to the degree of differential expression, represented by log_2_FoldChange value. The terms obtained as overrepresented after GSEA were considered significant if *p*-value was less than 0.05. Visualization of the results was achieved using enrichplot R package (version 4.4.1, R Core Team, Vienna, Austria) [81]. Gene Ontology [77] and Reactome-based overrepresentation analysis of DEGs was performed using the clusterProfiler R package (version 4.4.1, R Core Team, Vienna, Austria) [79] with gene length bias corrected. Kyoto Encyclopedia of Genes and Genomes database [82] was used when testing the statistical enrichment of DEGs in KEGG pathways, also carried out in clusterProfiler package (version 4.4.1, R Core Team, Vienna, Austria) [79]. GO and Reactome terms, as well as KEGG pathways with *p*-value less than 0.05, were considered significantly enriched by DEGs. For data visualization, ggplot2 R package (version 4.4.1, R Core Team, Vienna, Austria) was used [70]. The results obtained from different sources were cross-referenced, compared, and visualized.

### 4.8. QRT-PCR

To validate the transcriptome findings, qRT-PCR was performed for 9 selected genes on cDNA samples of 2.5 mM, 5 mM, and 10 mM Phe treatment groups (primers available upon request). qRT-PCR reactions were performed with 10 ng of cDNA using SYBR™ Green Universal Master Mix (Thermo Fisher Scientific, Waltham, MA, USA) and GAPDH as an endogenous control to normalize the obtained results. Reactions were carried out on 7500 Real-Time PCR system and all samples were run in triplicates. Relative quantification analysis was performed using comparative ΔΔCt method, with untreated cells as calibrator. 

### 4.9. Statistical Analysis

Statistical analysis was conducted to validate the obtained experimental results. As per MTT test, Welch’s t-test was applied to assess statistical significance of cell viability alterations, since variance was not homogenous upon Levene’s test. For PCA results, both with and without batch effect correction, clustering was statistically confirmed using one-way ANOVA followed by Tukey and Dunnett’s post hoc test, allowing comparison between treatment groups as well as with control cells. In differential expression analysis, Wald test was used and the resulting *p*-values were adjusted using the Benjamini and Hochberg’s method [78], in order to minimize the false discovery rate. For GSEA and ORA data analysis, Kolmogorov–Smirnov test and hypergeometric test were used for assessment of statistical significance, respectively. Lastly, the statistical significance of differential gene expression obtained by qRT-PCR was evaluated using Student’s t-test, after confirming the homogeneity of variances by Levene’s test (*p*-values > 0.05). Holm–Bonferroni correction [83] was applied, as it is more stringent than Benjamini–Hochberg method, ensuring less errors and greater reliability of statistical tests. In all cases, *p*-values or adjusted *p*-values less than 0.05 were considered statistically significant. All results were statistically analyzed using rstatix [84], DescTools [85], and clusterProfiler [79] R packages.

## Figures and Tables

**Figure 1 ijms-25-10019-f001:**
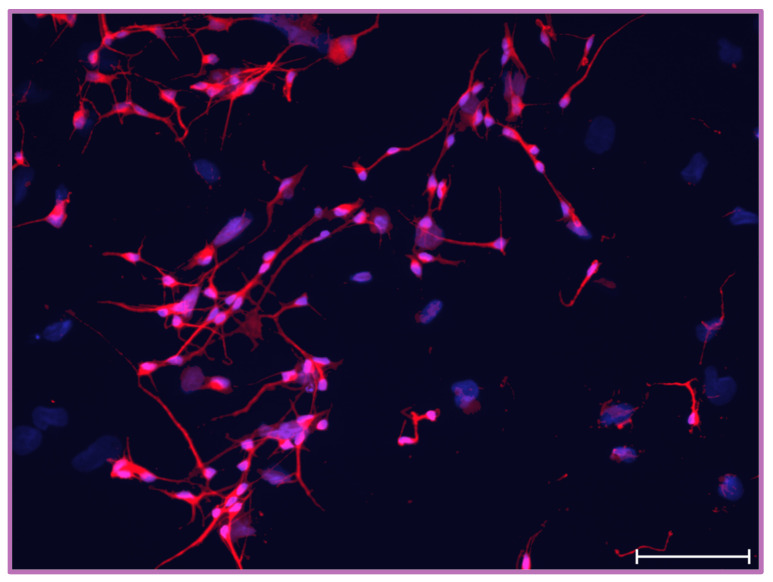
Immunostaining of NT2/N with the anti-MAP2 antibody (red). Mature neuronal phenotype of NT2-derived neurons was confirmed by observing MAP2, a neuron-specific cytoskeletal protein. Magnification used 20×, scale bar 100 μm.

**Figure 2 ijms-25-10019-f002:**
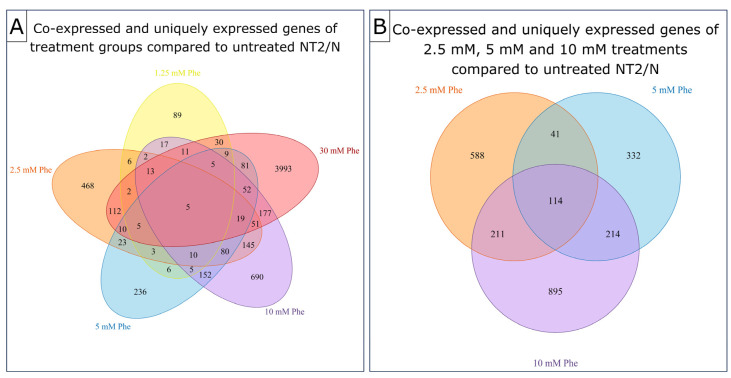
Gene co-expression between treatment groups in comparison to control cells (untreated NT2/N). Group treated with 1.25 mM Phe shows the lowest number of uniquely expressed genes, while group treated with 30 mM Phe has far more uniquely expressed genes compared to other groups (**A**). There are 5 genes co-expressed between all treatments (**A**). Groups treated with 2.5 mM, 5 mM, and 10 mM Phe show increased number of shared genes (**B**).

**Figure 3 ijms-25-10019-f003:**
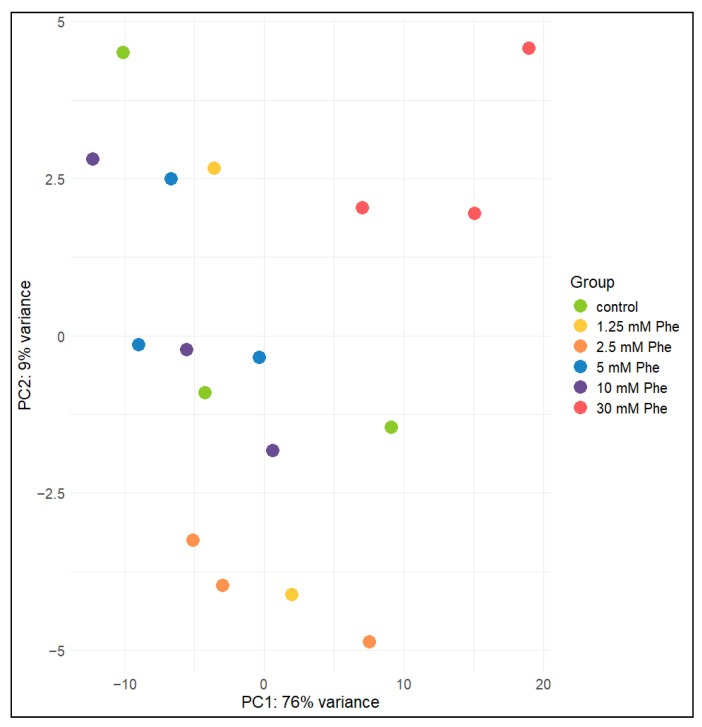
Principal component analysis biplot. PCA shows clear clustering of samples treated with 30 mM Phe. Samples treated with lower concentrations are clustered together, along with untreated cells. PCA points out the difference between highest Phe concentration group and the rest of the samples, explaining around 85% of the variability. Top 5 loadings contributing to PC1 are *CCND1*, *CPA4*, *RGS5*, *ADAMTS1*, and *CCND2*, while, for PC2, top 5 loadings are *FOSB*, *EGR1*, *FOS*, *COL4A2*, and *NES* (Appendix A).

**Figure 4 ijms-25-10019-f004:**
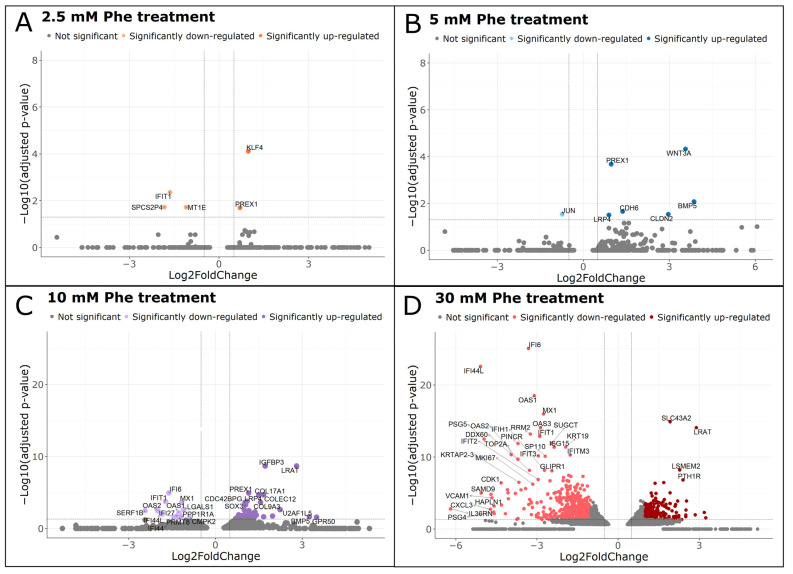
Significantly up- and down-regulated DEGs observed for each treatment group in comparison to untreated NT2/N. Genes with significantly altered expression upon exposure to 2.5 mM (**A**), 5 mM (**B**), 10 mM (**C**), and 30 mM (**D**) Phe visualized on volcano plots. Horizontal dashed lines represent adjusted *p*-value cutoff (0.05) on a logarithmic scale for base 10, while vertical dashed lines represent absolute log_2_FoldChange cutoff (0.5). Dots visualized in lighter colors indicate down-regulated genes, while darker-colored dots indicate up-regulated genes. If possible, all DEGs were shown on volcano plots (**A**,**B**), and, if not, at least top 20 DEGs were represented on the plot (**C**,**D**).

**Figure 5 ijms-25-10019-f005:**
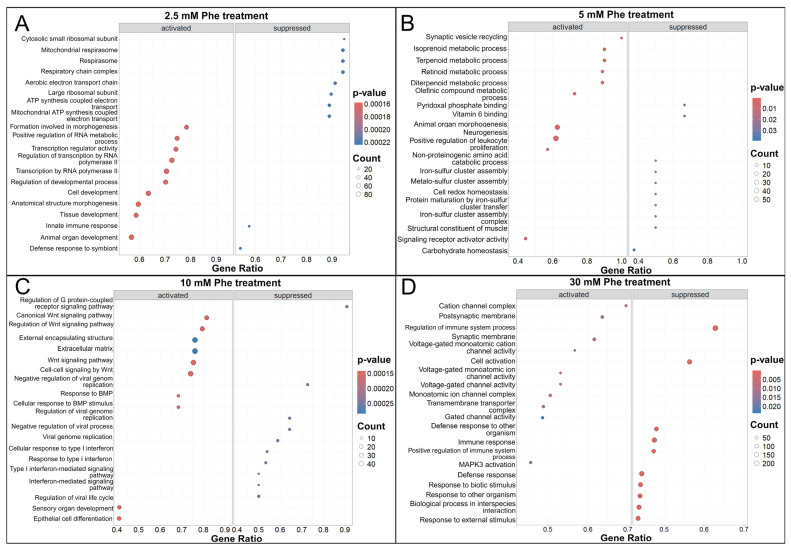
Significant GO terms obtained by GSEA of 2.5 mM, 5 mM, 10 mM, and 30 mM Phe treatments in comparison to control group. Upon GSEA for 2.5 mM treatment, GO terms that emerged include G protein-coupled receptor signaling pathway, actin cytoskeleton organization, and neuron projection morphogenesis, along with terms connected to mitochondrial respirasome and ATP synthesis (**A**). In 5 mM Phe-treated group, among enriched GO terms were actin filament polymerization, synaptic vesicle recycling, and synaptic signaling (**B**). In NT2/N treated with 10 mM Phe, overrepresented GO terms include neurotransmitter loading into synaptic vesicle, neuron projection guidance, and axonogenesis (**C**). In the group treated with highest Phe concentration (30 mM), enriched terms were cell migration, cell adhesion, and response to external biotic stimulus, but also actin cytoskeleton and neuron to neuron synapse (**D**). Dotplots represent the top 20 most overrepresented GO terms for each treatment group.

**Figure 6 ijms-25-10019-f006:**
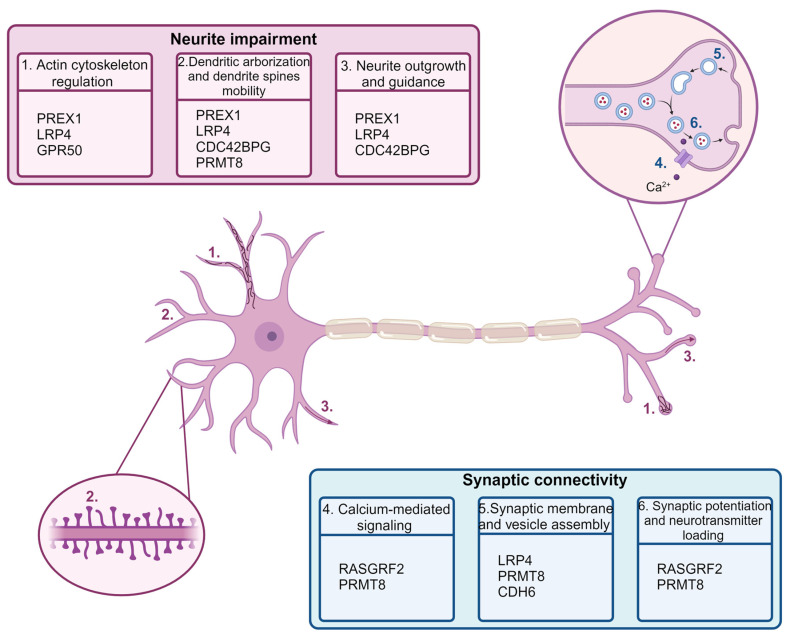
DEGs observed upon treatment with 2.5 mM, 5 mM, and 10 mM Phe and their known roles in human neuron dysfunction. Two distinct processes emerged as important in Phe-mediated effect on neurons: neurite impairment and synaptic connectivity. Neurite impairment due to Phe exposure is observed in gene expression alterations contributing to actin cytoskeleton regulation, dendritic arborization and dendritic spines mobility, and neurite outgrowth and guidance. Phe-mediated effect on synaptic connectivity was observed as gene expression changes connected to calcium-mediated signaling, synaptic membrane and vesicle assembly, and synaptic potentiation and neurotransmitter loading. Created with BioRender.com (accessed on 13 August 2024).

**Table 1 ijms-25-10019-t001:** Statistically significant DEGs identified in sample groups treated with 2.5 mM, 5 mM, and 10 mM Phe with roles in central nervous system and connection to neuronal functioning and diseases. Asterisks are indicating statistical significance as follows: * adjusted *p*-value < 0.05, ** adjusted *p*-value < 0.01, *** adjusted *p*-value < 0.001, **** adjusted *p*-value < 0.0001.

Treatment Group	Gene Name	Function	Differential Expression	Fold Change
2.5 mM Phe	*KLF4*	Transcription factor	Up-regulated	1.98 ****
	*MT1E*	Metal-binding protein	Down-regulated	0.47 **
	*PREX1*	Signaling protein (guanine nucleotide exchange factor)	Up-regulated	1.63 *
5 mM Phe	*WNT3A*	Signaling protein	Up-regulated	11.88 ****
	*PREX1*	Signaling protein (guanine nucleotide exchange factor)	Up-regulated	1.96 ***
	*CDH6*	Cell adhesion protein	Up-regulated	2.60 *
	*LRP4*	Low-density lipoprotein receptor-related protein	Up-regulated	1.86 *
10 mM Phe	*IGFBP3*	Binding protein (transcription regulator)	Up-regulated	3.29 ****
	*PREX1*	Signaling protein (guanine nucleotide exchange factor)	Up-regulated	2.22 ****
	*LRP4*	Low-density lipoprotein receptor-related protein	Up-regulated	2.10 ***
	*CDC42BPG*	Enzyme (protein kinase)	Up-regulated	2.03 ***
	*SOX3*	Transcription factor	Up-regulated	2.06 ***
	*CDH6*	Cell adhesion protein	Up-regulated	2.24 **
	*AEBP1*	Enzyme (carboxypeptidase)	Up-regulated	2.24 **
	*PPP1R13L*	Enzyme (protein phosphatase)	Up-regulated	2.10 *
	*PCSK9*	Enzyme (subtilisin-like proprotein convertase)	Up-regulated	2.66 *
	*NTNG2*	Cell adhesion protein	Up-regulated	1.83 *
	*GPR50*	G-protein coupled receptor	Up-regulated	11.31 *
	*RASGRF2*	Nucleotide exchange factor	Up-regulated	1.89 *
	*PRMT8*	Enzyme (arginine methyltransferase)	Down-regulated	0.43 *

## Data Availability

Raw genetic data are generated and stored at the Institute of Molecular Genetics and Genetic Engineering, University of Belgrade, and are available from the corresponding author upon reasonable request.

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
