# Peer review of "Transcriptome Profiling of Phenylalanine-Treated Human Neuronal Model: Spotlight on Neurite Impairment and Synaptic Connectivity"

_ijms, 2024, doi:10.3390/ijms251810019_

Round 1
Reviewer 1 Report
Comments and Suggestions for Authors
This study investigates the transcriptome alterations after Phe exposure of human neuronal model, which authors claim to be the first study that strives to evaluate the effect of Phe on gene expression in mature neuronal model. I would like to congratulate the authors for an amazing study which provides path for new research lines into novel therapeutics options as it elucidates some molecular mechanism in this complex PKU pathology. Overall, the manuscript is very nicely written, very clear and detailed.
I only have few comments and discussion points:
- Introduction seems a bit too long, it reads more as a review; e.g. it is not needed to have a full paragraph on animal models or 3D organoids as these are not the focus of the study.
- Regarding the PCA - could it be that there is any batch effect present? Either during sequencing or sample preparation? This could potentially indicate some minor clustering alongside y-axis.
- Was DESeq2 performed on raw counts or FPKM (as it is not mentioned)?
- As GSEA usually outperforms ORA, I would suggest to keep just GSEA results in the main text and more ORA results in supplementary data, and only comment the differences/similarities of the two analysis within the main text. Another suggestion would also be to move the KEGG results as supplementary data, and also since you are working with human samples, you could also check the Reactome pathways database as an additional complementary pathway analysis.
- Even though a bit long, the discussion is written nicely, however, could you please add the paragraph on what do you see are the limitations of your study.
Minor details:
- Figure 1: add rule for scale information on the photo
- Figures S1/S2: increase font size
- Figure 2: increase readability of the text (bit blurry once you zoom in)
- Table 1: as mentioned in the text, perhaps it would be nice to add a column with brief description of a gene function
- Figure 5: increase readability; I would suggest to keep only the top terms and provide the rest as supplementary tables (in the figure legend it says "top 20 terms are depicted", however, more terms are actually in the figures)
- Perhaps it would be good to have a short and concise overview of the whole study design
Comments on the Quality of English LanguageEnglish language is fine, very clear and easy to follow.
Reviewer 2 Report
Comments and Suggestions for Authors
Dear authors, your research is very interesting and has good contributions in this area of ​​knowledge. However, before publishing it, you need to address this series of points.
1) It is important that you include a section on Statistical Analysis (Materials and Methods) and explain in an orderly manner the tests you applied and their justification.
2) Figure 2, Why is the control group not shown?
3) Figure 3, the PCA shows the multivariate ordination. However, the x and y axes are explained by a series of variables in their linear combination, which are not indicated in the graph or the figure caption. Explain and include them in the figure caption.
4) The PCA is an ordination technique and not an inferential one, therefore, it cannot be indicated that they are different groups only with the PCA. An inferential statistical test is required to contrast the set of hypotheses. Include it and justify why it is applied. This point is very delicate because the n of the “groups” is very small and the one-way ANOVA using the score factors of PC1 and PC2 as a response variable would perhaps provide a contrast. However, there will be a bias due to the sample size of the groups (explain and justify).
5) Figure 4, the font size on the x and y axes are very small, adjust. Also, in A and B, the y-axis, only adjusts up to the value of 7. In the case of C and D, adjust up to the value of 30.
6) The labels in C and D are very small, adjust.
7) Table 1, the adjusted probability values, apply the international rule:
NS = P>0.05, *P<0.05, **P<0.01, ***P<0.001, ****P<0.0001, and so on as the case may be, although four asterisks are clear, the effect of contrast. Adjust and eliminate the last column of the Table.
8) The font size on both axes of the graphs is not legible, adjust and homogenize the size of the symbols.
9) Same situation in Figure 6, in addition, the titles in some of the categories overlap. Adjust.
10) Same as Figure 7.
Best regards,
Moderate editing of English language required.
Round 2
Reviewer 2 Report
Comments and Suggestions for Authors
Dear authors,
You are very kind in making the adjustments and changes to your manuscript. There is only one comment left: indicate whether you verified the homogeneity of variances for the Student t test. In the ANOVA, it is not required because you analyzed the factor scores of the PCA, which standardizes the response variable. But in the t-test, this does not happen. Please clarify and indicate.
Kind regards,
Comments on the Quality of English LanguageMinor editing of English language required.
Author Response
Comment 1: You are very kind in making the adjustments and changes to your manuscript. There is only one comment left: indicate whether you verified the homogeneity of variances for the Student t test. In the ANOVA, it is not required because you analyzed the factor scores of the PCA, which standardizes the response variable. But in the t-test, this does not happen. Please clarify and indicate.
Response 1: We would like to thank the Reviewer for this comment. As per qRT-PCR, we have performed Levene’s test to assess homogeneity of variances before Student’s t-test and concluded that for all genes tested, variances were homogenous. However, we realized that we did not include this information in the main text, which we now corrected (section 4.9. Statistical Analysis, page 19, lines 712-714), and therefore we appreciate this being highlighted by the Reviewer. We would also like to thank the Reviewer for drawing our attention to this matter in the case of MTT test, since we have now revised our methods applied for statistical analysis upon MTT test and now used Welch’s t-test (section 4.9. Statistical Analysis, page 18, lines 703-705). We opted for this test because it still assumes normal distribution of the data (which we confirmed by Shapiro-Wilk test), but does not assume equal variances. Upon Welch’s t-test the p-values did not vary markedly, compared to previously performed Student’s t-test, and therefore, the results (including asterisks on the Figure S2) did not change.
Response to Comments on the Quality of English Language
Point 1: Minor editing of English language required.
Response 1: We would like to thank the Reviewer for this comment, and we revised the manuscript accordingly.
Round 3
Reviewer 2 Report
Comments and Suggestions for Authors
Dear author,
It was very kind of you to make the adjustment to your manuscript.
Kind regards,
Comments on the Quality of English LanguageMinor editing of English language required.